# LASER: ATTENTION WITH EXPONENTIAL TRANSFORMATION

## ABSTRACT

Transformers have had tremendous impact for several sequence related tasks. The Transformer's ability to retrieve from any part of the sequence via a parameterized query-key-value mechanism - Softmax based dot-product attention The Softmax based dot-product attention mechanism plays a key role in t. However, the softmax operation can backpropagate small gradients thus inhibiting learning. In this paper, we fix this by introducing a new attention mechanism called LASER Attention, which admits a log-sum-exp structure and propagates a larger gradient signal. We show that LASER Attention can be implemented by making small modifications to existing attention implementations. We conduct experiments on large language models (LLMs) with upto 2.2 billion parameters where we show improvements of upto 3.38% and ∼1% on an average compared to standard attention on downstream one-shot evaluations. We also evaluate on transformers spanning different modalities (vision, speech and text): Vision Transformer (ViT) on Imagenet (1.2% improvement in accuracy), Conformer on the Librispeech speech-to-text task (2.25% relative improvement) and encoder-only BERT Transformer with 2.2 billion parameters (0.93% relative improvement).

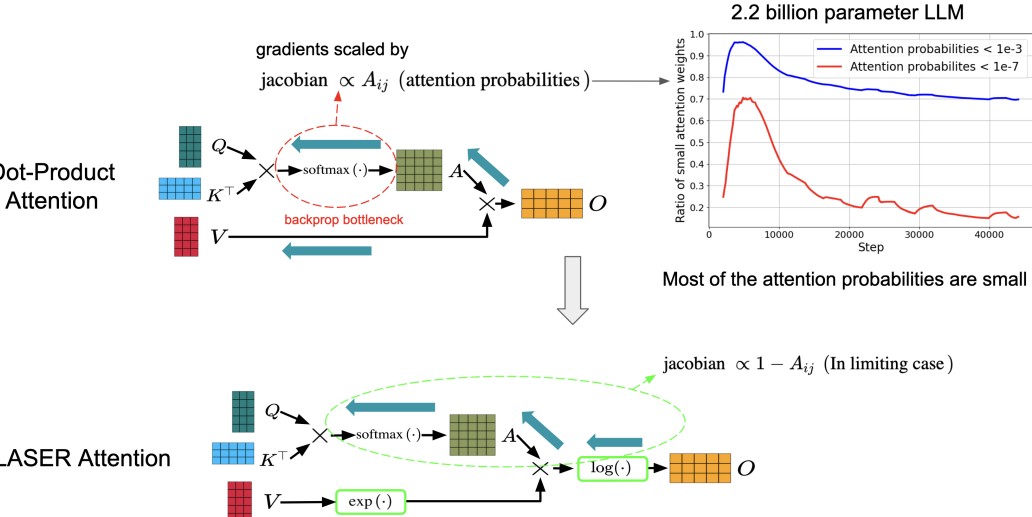

Figure 1: Backpropagating gradients through the softmax operation in attention mechanism requires scaling with Jacobian of softmax. We show that this Jacobian is proportional to the magnitude of attention weights, which are typically small in large language models (LLMs) with about 80% of the weights less than $10^{-3}$ and about 20% less than $10^{-7}$. We implement a fix called LASER attention that involves conducting Dot-Product Attention with an $\exp(\cdot)$-transformed value matrix $V$, i.e., conducting attention on $\exp(V)$. We show that LASER admits a larger Jacobian, easy to implement and does not require any change to the underlying attention function, which may have a more nuanced implementation (e.g., FlashAttention (Dao et al., 2022)). In the image, $\exp(.)$ and $\log(.)$ are element-wise transformations.

## 1 INTRODUCTION

In recent years, the transformer architecture and its associated attention mechanisms have gained prominence over traditional models like LSTMs for various sequence-based tasks due to their ability to better capture long-range dependencies without suffering from the vanishing gradient problem. The key component of Transformer, Attention mechanism, assigns different weights to previous tokens in a sequence, indicating their relative importance, and these weights are computed via a softmax function (Vaswani et al., 2017). The transformer architecture consists of multiple stacked layers, where each layer operates on the output of the previous one, forming the transformer encoder or decoder. Learning within the transformer is performed via gradient backpropagation, wherein gradients propagate backward through the network layer by layer using the chain rule (LeCun et al., 2002). However, as gradients backpropagate through multiple layers, their magnitude tends to diminish, resulting in weaker gradients reaching the bottom layers. This phenomenon can hinder effective learning in deeper layers. Residual connections (He et al., 2016) are used in Transformers to bypass the layers via skip connections to avoid this issue. However, it behooves us to develop layers which backpropagate gradients effectively.

In this paper we identify a similar gradient vanishing issue in attention mechanism of Transformer. We show that the softmax operation backpropagate small gradients in large language models. Based on this observation, we propose a small modification to attention mechanism - LASER attention (LogArithm of Summed Exponentials of Representations). LASER is equivalent to conducting attention on exponentially transformed inputs and takes a log-sum-exp structure. We analytically show that gradients propagated via LASER attention are typically large. Since $\exp(\cdot)$ transformation in LASER can lead to overflows during implementation, we develop a novel implementation - Log-Weighted-Sum-Exp trick, inspired from Log-Sum-Exp trick (Blanchard et al., 2019). This technique allows LASER to scale to large models with upto 2.2 billion parameter models. We show that our implementation requires small modifications, and doesn't need any changes to underlying attention mechanisms which may or may not admit more nuanced implementation (for e.g., FlashAttention).

We conduct thorough empirical verification across a variety of transformer models, including Conformer for Librispeech speech-to-text (Gulati et al., 2020), Vision Transformer(Dosovitskiy et al., 2021) for ImageNet classification (Deng et al., 2009), decoder-only text transformer (Brown et al., 2020) on C4 dataset (Raffel et al., 2020) and encoder-only BERT ((Devlin et al., 2018)). We conduct experiments on decoder-only causal language models from 234 million parameters to 2.2 billion parameter models, where we demonstate improvements of up to 1.7% relative improvement in test loss over standard attention function. We conduct one-shot evaluation on 17 downstream tasks and show that LASER outperforms Standard attention on 14 tasks with upto 3.38% difference in accuracy and an average of 1% accuracy difference. On BERT with 2.2 billion parameter we show a relative improvement of 0.93% on masked language modeling prediction error rate. We also show a 4.67% relative improvement in validation error rate in Vision Transformer and 1.2% absolute improvement in accuracy, and a 2.25% relative improvement in validation word error rate in the Conformer benchmark. Furthermore, LASER attention can be implemented with small modifications to the inputs of traditional attention mechanism, making it a feasible and effective enhancement for a wide range of Transformer architectures.

## 2 RELATED WORK

Attention mechanism was used in Bahdanau et al. (2015) to drastically improve machine translation performance compared to encoder-decoder recurrent neural networks (RNNs) (Cho, 2014). This was later adopted in Transformers (Vaswani et al., 2017), which introduced self-attention to improve the performance in machine translation even further. Efficient attention mechanisms have been an active area of research due to the quadratic complexity in sequence length of Attention, which prevents long-context language modeling. One notable contribution is Linear Attention (Katharopoulos et al., 2020), which reduces the quadratic complexity of self-attention to linear by approximating the softmax function. Similarly, the Performer (Choromanski et al., 2021) uses kernel methods to achieve linear complexity in transformers, making them more scalable for large-scale applications while retaining competitive performance in various tasks.

The Mamba architecture, particularly Mamba-2, introduces state-space models (SSMs) as a replacement for traditional attention. Models like SSD from Mamba-2 (Dao & Gu, 2024) and S6 (S4+selection+scan) (Gu & Dao, 2023) showcase an efficient way to model long-range dependencies without the use of attention, leading to faster computation. However, despite these innovations, attention-based models like LLaMA 3 (Dubey et al., 2024) continue to dominate large-scale applications, particularly through advancements in context parallelism, which ensures scalability while maintaining the strengths of attention mechanisms in transformer models.

Efficient attention mechanisms have become critical in handling large-scale data and long sequences, especially in transformer-based architectures. FlashAttention (Dao et al., 2022; 2024), is a recent advancement that optimizes memory and computational speed by improving memory bandwidth utilization during attention computation, making it both fast and memory-efficient. This mechanism is used for faster inference and training, particularly when scaling up to large sequence lengths. Routing Transformers take a different approach by introducing a mechanism that sparsifies attention through dynamic routing, where only the most relevant tokens are attended to during each attention step (Roy et al., 2021) with subquadratic computational complexity in sequence length. Similarly, Longformer (Beltagy et al., 2020) modifies the standard self-attention mechanism to handle long documents by combining local attention with selected global attention tokens. Sparse Transformers (Child et al., 2019) use fixed sparse attention patterns, enabling them to efficiently handle very long sequences by reducing the quadratic complexity of standard attention to linear or sub-quadratic in practice. By focusing only on a sparse subset of the tokens in each layer. LASER Attention can be thought of as complementing these approaches, as it conducts attention using the exponential transformation of inputs, without any change to underlying attention function.

## 3 LASER ATTENTION- LOGARITHM OF SUMMED EXPONENTIALS OF REPRESENTATIONS

We first formally introduce the standard softmax dot-product attention used in Transformers (Vaswani et al., 2017) in Section 3.1. In Section 3.2, we introduce LASER Attention by first deriving the gradients of standard attention by considering a simple case of sequence length 2.

### 3.1 TRANSFORMERS AND SOFTMAX DOT-PRODUCT ATTENTION

Let $X \in \mathbb{R}^{N \times d}$ be the input representing the sequence with $N$ tokens where the rows are representations of the tokens. Let $A : \mathbb{R}^{N \times d} \to \mathbb{R}^{N \times d}$ denote the attention function. Attention function is the only operation in the transformer which is applied across the sequence axis. We describe the transformer layer $T_l : \mathbb{R}^{N \times d} \to \mathbb{R}^{N \times d}$ similar to Katharopoulos et al. (2020) as follows:

$$T_l(X) = f_l(X + A_l(X)W_O).$$

Here $f_l : \mathbb{R}^{N \times d} \to \mathbb{R}^{N \times d}$ is usually implemented using a 2-layer feed-forward neural network which acts on each token representation independently and $W_O \in \mathbb{R}^{d \times d}$ is a tunable parameter matrix. A single head attention mechanism (Vaswani et al., 2017) can be described as follows:

$$K = XW_K^{(l)} \in \mathbb{R}^{N \times d}, \quad Q = XW_Q^{(l)} \in \mathbb{R}^{N \times d}, \quad V = XW_V^{(l)} \in \mathbb{R}^{N \times d},$$
$$A_l(X) = \text{softmax}(QK^\top)V.$$

The $\text{softmax}$ (Bridle, 1990) operation is applied row-wise. Layer normalizations (Ba et al., 2016) are applied before $f_l(.)$, and $A_l(.)$, but we omit this for brevity. A transformer comprises of stacking functions $T_l(X), l \in \{1, \ldots, L\}$ sandwiched by embedding layer $E : \mathbb{R}^{N \times V} \to \mathbb{R}^{N \times d}$ and softmax layer $S : \mathbb{R}^{N \times d} \to \mathbb{R}^{N \times V}$ as follows:

$$T(X) = S \circ T_L \circ \cdots \circ T_1 \circ E(X) \in \mathbb{R}^{N \times V},$$

where the inputs $X \in \mathbb{R}^{N \times V}$. Let $\ell(T(X), Y)$ be the loss function used to learn the parameters of the transformer $T$, where $Y$ represents label information. Autoregressive language modeling (Radford et al., 2018; Brown et al., 2020) involves using a causal mask $M$ which is lower triangular

and is multiplied before the softmax operation as follows:

$$A_l(X) = \text{softmax}(M \odot QK^\top)V,$$
$$M_{ij} = 1 \quad \text{if } i \leq j,$$
$$= 0 \quad \text{else,}$$

where $\odot$ denotes element-wise multiplication. During training, gradients $\frac{\partial \ell}{\partial W_K}$, $\frac{\partial \ell}{\partial W_Q}$, $\frac{\partial \ell}{\partial W_V}$ are computed via backpropagation in a layer-by-layer fashion from layer-$L$ to layer-1 and are used to update the parameters. In the next section we analyze the gradients for a simple case of sequence length $N = 2$ and similarly analyze LASER attention.

### 3.2 GRADIENT ANALYSIS OF ATTENTION

For simplicity, let the sequence length $N = 2$ with attention weights $A = \text{softmax}(KQ^\top)$ and attention logits as $\tilde{A} = KQ^\top$. If we expand the matrices $A$ and $\tilde{A}$, we get the following:

$$
\begin{pmatrix} a_{11} & a_{12} \\ a_{21} & a_{22} \end{pmatrix} = \text{softmax} \begin{pmatrix} \tilde{a}_{11} & \tilde{a}_{12} \\ \tilde{a}_{21} & \tilde{a}_{22} \end{pmatrix}
$$
$$
= \begin{pmatrix} \frac{\exp(\tilde{a}_{11})}{\exp(\tilde{a}_{11})+\exp(\tilde{a}_{12})} & \frac{\exp(\tilde{a}_{12})}{\exp(\tilde{a}_{11})+\exp(\tilde{a}_{12})} \\ \frac{\exp(\tilde{a}_{21})}{\exp(\tilde{a}_{21})+\exp(\tilde{a}_{22})} & \frac{\exp(\tilde{a}_{22})}{\exp(\tilde{a}_{21})+\exp(\tilde{a}_{22})} \end{pmatrix}
$$
$$
= \begin{pmatrix} \sigma(\tilde{a}_{11} - \tilde{a}_{12}) & 1 - \sigma(\tilde{a}_{11} - \tilde{a}_{12}) \\ \sigma(\tilde{a}_{21} - \tilde{a}_{22}) & 1 - \sigma(\tilde{a}_{21} - \tilde{a}_{22}) \end{pmatrix}, \tag{1}
$$

where $\sigma$ denotes the sigmoid operation $\sigma(x) = 1/(1 + \exp(-x))$. Let the representation dimension $d = 1$, then the attention result will be as follows:

$$
\text{Attention output:} \quad A_l(X) = \begin{pmatrix} o_1 \\ o_2 \end{pmatrix} = \begin{pmatrix} \sigma(\tilde{a}_{11} - \tilde{a}_{12})v_1 + (1 - \sigma(\tilde{a}_{11} - \tilde{a}_{12}))v_2 \\ \sigma(\tilde{a}_{21} - \tilde{a}_{22})v_1 + (1 - \sigma(\tilde{a}_{21} - \tilde{a}_{22}))v_2, \end{pmatrix} \tag{2}
$$

where $V = \begin{pmatrix} v_1 \\ v_2 \end{pmatrix}$. To compute the gradient with respect to $\tilde{A}$, we can use chain rule:

$$
\underbrace{\frac{\partial \ell}{\partial \tilde{A}}}_{\text{gradient backpropagated}} = \frac{\partial \ell}{\partial A_l(X)} \cdot \underbrace{\frac{\partial A_l(X)}{\partial \tilde{A}}}_{\text{Jacobian}}
$$

If the Jacobian is small in magnitude then the gradient backpropagated will also be small. We now analyze an element of the Jacobian:

$$
\text{Attention Jacobian:} \quad \frac{\partial o_1}{\partial \tilde{a}_{11}} = v_1 \sigma(\tilde{a}_{11} - \tilde{a}_{12})(1 - \sigma(\tilde{a}_{11} - \tilde{a}_{12})) - v_2 \sigma(\tilde{a}_{11} - \tilde{a}_{12})(1 - \sigma(\tilde{a}_{11} - \tilde{a}_{12}))
$$
$$
= (v_1 - v_2) \underbrace{\sigma(\tilde{a}_{11} - \tilde{a}_{12})(1 - \sigma(\tilde{a}_{11} - \tilde{a}_{12}))}_{\text{possible saturation}} \tag{3}
$$

The sigmoid function value, $\sigma(\tilde{a}_{11} - \tilde{a}_{12})$ saturates to 1 when $\tilde{a}_{11} - \tilde{a}_{12}$ becomes sufficiently large. Conversely, when $\tilde{a}_{11} - \tilde{a}_{12}$ is large and negative, the function value saturates to 0. In both cases, saturation leads to vanishing gradients, where the gradient becomes very small. This phenomenon is a well-documented limitation of the sigmoid function (LeCun et al., 2002).

We extend this observation to sequence length of size $N$ as follows:

**Lemma 3.1** (Gradient saturation in softmax). *Let $a \in \mathbb{R}^N$ be a row in attention weights/probabilities $A$ and similarly let $\tilde{a}$ be a row in attention logits $\tilde{A}$, then:*

$$a = \text{softmax}(\tilde{a})$$
$$\frac{\partial \ell}{\partial \tilde{a}} = (\text{diag}(a) - aa^\top)\frac{\partial \ell}{\partial a}$$

We give a proof of this lemma in Section A.2

> **Key Observation.** From Lemma 3.1, it can be seen that the Jacobian of softmax operation is proportional to attention probabilities. For a 2.2 billion parameter autoregressive language model, we observe (see Figure 1) that about 80% of attention probabilities are less than $10^{-3}$ and about 20% are less than $10^{-7}$, during pretraining. Thus gradient backpropagated through softmax operation is scaled by very small values.

To address this issue, we now introduce LASER Attention which applies attention on $\exp(V)$, elementwise exponentials of value matrix $V$ as follows:

$$\exp(A_l(X)) = \text{softmax}(QK^\top)\exp(V)$$
$$A_l(X) = \log(\text{softmax}(QK^\top)\exp(V)) \quad \rightarrow \quad \text{LASER Attention} \tag{4}$$

where $\log(.)$ is applied elementwise. Expanding (4) for $N = 2$ and $d = 1$ as done for standard attention in (2) gives:

$$\text{LASER output:} \quad \begin{pmatrix} o_1 \\ o_2 \end{pmatrix} = \begin{pmatrix} \log(\sigma(\tilde{a}_{11} - \tilde{a}_{12})\exp(v_1) + (1 - \sigma(\tilde{a}_{11} - \tilde{a}_{12}))\exp(v_2)) \\ \log(\sigma(\tilde{a}_{21} - \tilde{a}_{22})\exp(v_1) + (1 - \sigma(\tilde{a}_{21} - \tilde{a}_{22}))\exp(v_2)), \end{pmatrix} \tag{5}$$

**Low gradient saturation.** Computing an element in the Jacobian $\frac{\partial A_l(X)}{\partial X}$ as done in (3) will give the following:

$$\text{LASER Jacobian:} \quad \frac{\partial o_1}{\partial \tilde{a}_{11}} = \frac{(\exp(v_1) - \exp(v_2))\sigma(\tilde{a}_{11} - \tilde{a}_{12})(1 - \sigma(\tilde{a}_{11} - \tilde{a}_{12}))}{\sigma(\tilde{a}_{11} - \tilde{a}_{12})\exp(v_1) + (1 - \sigma(\tilde{a}_{11} - \tilde{a}_{12}))\exp(v_2)}$$
$$= \frac{(\exp(v_1) - \exp(v_2))\sigma(\tilde{a}_{11} - \tilde{a}_{12})(1 - \sigma(\tilde{a}_{11} - \tilde{a}_{12}))}{\sigma(\tilde{a}_{11} - \tilde{a}_{12})(\exp(v_1) - \exp(v_2)) + \exp(v_2)} \tag{6}$$

Without loss of generality, if $v_1 \gg v_2$, then

$$\text{LASER Jacobian:} \quad \frac{\partial o_1}{\partial \tilde{a}_{11}} = \frac{\sigma(\tilde{a}_{11} - \tilde{a}_{12})(1 - \sigma(\tilde{a}_{11} - \tilde{a}_{12}))}{\sigma(\tilde{a}_{11} - \tilde{a}_{12}) + \exp(v_2)/(\exp(v_1) - \exp(v_2))}$$
$$\approx \underbrace{(1 - \sigma(\tilde{a}_{11} - \tilde{a}_{12}))}_{\text{low saturation}}$$

The approximation is due to $\exp(v_2)/(\exp(v_1) - \exp(v_2) \approx 0$.

**Relation between LASER attention and max function.** From 1 and (5), LASER output can be written in a log-sum-exp form (Blanchard et al., 2019) as follows:

$$o_1 = \log(a_{11}\exp(v_1) + a_{12}\exp(v_2))$$
$$= \log(\exp(v_1 + \log(a_{11}) + \exp(v_2 + \log(a_{12})) \tag{7}$$

Log-exp-sum function can be thought of as a differentiable approximation of $\max$ function:

**Lemma 3.2** (Boyd & Vandenberghe (2004))**.** *The function $f(x_1, \ldots, x_n) = \log(e^{x_1} + \cdots + e^{x_n})$ is convex on $\mathbb{R}^n$. This function can be interpreted as a differentiable approximation of the max function, since*

$$\max\{x_1, \ldots, x_n\} \leq f(x_1, \ldots, x_n) \leq \max\{x_1, \ldots, x_n\} + \log n$$

*for all $x \in \mathbb{R}^n$. (The second inequality is tight when all components of $x$ are equal.)*

Given that $\max(x_1, \ldots, x_n)$ function is not differentiable at points where two or more elements take the same value, log-sum-exp can serve as a differentiable approximation. Using Lemma 3.2, we can relate LASER (7) to $\max(\cdot)$ operation as follows:

$$\max(v_1 + \log(a_{11}), v_2 + \log(a_{12})) \leq o_1 \leq \max(v_1 + \log(a_{11}), v_2 + \log(a_{12})) + \log(2)$$

### 3.3 LASER IMPLEMENTATION VIA LOG-WEIGHTED-SUM-EXP TRICK

In this section we explore implementing LASER and provide a pseudocode. Given the log-sum-exp structure from (7):

$$o_1 = \log(\sigma(\tilde{a}_{11} - \tilde{a}_{12}) \exp(v_1) + (1 - \sigma(\tilde{a}_{11} - \tilde{a}_{12})) \exp(v_2)),$$

one can notice that $\exp(.)$ operations can lead to overflow. This problem has been recognized in Blanchard et al. (2019) and "Log-sum-exp trick" is used to avoid overflows. However, the log-sum-exp trick cannot be applied directly as it would be difficult to implement without changing the underlying attention function. We propose a "Log-weighted-sum trick", where we subtract the maximum value $m = \max(v_1, v_2)$ from $v_1$ and $v_2$ and rewrite the above equation as follows:

$$o_1 = \log((\sigma(\tilde{a}_{11} - \tilde{a}_{12}) \exp(v_1 - m) + (1 - \sigma(\tilde{a}_{11} - \tilde{a}_{12})) \exp(v_2 - m)) * \exp(m))$$
$$= \log(\sigma(\tilde{a}_{11} - \tilde{a}_{12}) \exp(v_1 - m) + (1 - \sigma(\tilde{a}_{11} - \tilde{a}_{12})) \exp(v_2 - m)) + m$$

Now conducting $\exp(.)$ operation on $v_1 - m$ and $v_2 - m$ will not lead to overflows. We can extend this to matrix-version (4) by conducting column-wise maximum of value matrix $V \in \mathbb{R}^{N \times d}$ as follows:

$$m_j = \max_{i \in \{1, \ldots, N\}} V_{ij}, \; j \in \{1, \ldots, d\}$$
$$\text{Define } \hat{V} \in \mathbb{R}^{N \times d} \text{ such that } \hat{V}_{ij} = (V_{ij} - m_j)$$

The above operations helps us conduct $\exp(.)$ operation without overflows. Then the final LASER attention operation would be as follows:

$$\text{Define } O \in \mathbb{R}^{N \times d} \text{ as: } O = \log(\text{softmax}(QK^\top) \exp(\hat{V}) \, \text{diag}(\exp(m)))$$
$$O_{ij} = (\log(\text{softmax}(QK^\top) \exp(\hat{V})))_{ij} + m_j,$$

Here, $m = (m_1, \ldots, m_d)$ and $\text{diag}(m)$ is a diagonal matrix with elements of $m$ as diagonals. The main use of our Log-weighted-sum-exp trick is, it allows us to implement LASER attention via merely modifying inputs and outputs of standard attention, without changing the underlying attention function. We show this in the following JAX (Bradbury et al., 2018) code, where we can implement LASER attention using standard attention functions.

Listing 1: JAX implementation of LASER attention requres a small change to existing attention implementations

```
# given key (B,N,H,S), value (B,N,H,S), query (B,N,H,S)
# B - batch size, N - sequence length, H - number of attention heads
# S - size of the head
m = jnp.max(value, axis=1, keepdims=True)  # max along sequence dimension
m = jax.lax.stop_gradient(m)  # stop the gradients along m
exp_value = jnp.exp(value - m)  # shifting the values
f = standard_attention  # Efficient attention - FlashAttention, etc.
attention_out = f(key, query, exp_value)
out = jnp.log(attention_out) + m  # adding back the max values
```

---

**Algorithm 1** LASER Attention with Log-Weighted-Sum-Exp Trick

---

1: **Input:** Values $V \in \mathbb{R}^{N \times d}$, Queries $Q \in \mathbb{R}^{N \times d}$, Keys $K \in \mathbb{R}^{N \times d}$
2: **Output:** LASER Attention output $O \in \mathbb{R}^{N \times d}$
3: Compute the column-wise maximum for the value matrix $V$

$$m_j = \max_{i \in \{1,\ldots,N\}} V_{ij}, \quad j \in \{1,\ldots,d\}$$

4: Subtract $m_j$ from each element of $V$

$$\hat{V} \in \mathbb{R}^{N \times d} \text{ such that } \hat{V}_{ij} = (V_{ij} - m_j) \quad \text{// Shift values to avoid overflow in } \exp(.)$$

5: Apply attention with Queries $Q$, Keys $K$ and Values $V$ with $m_j, j \in \{1,\ldots,d\}$ added back to the output

$$\text{Define } O \in \mathbb{R}^{N \times d} \text{ as: } (O)_{ij} = (\log(\text{softmax}(QK^\top)\exp(\hat{V})))_{ij} + m_j$$

6: **return** LASER attention output $O$

---

## 4 EXPERIMENTAL RESULTS

### 4.1 AUTOREGRESSIVE LANGUAGE MODELING ON C4

In this section, we compare the performance of LASER Attention with standard attention mechanisms in the context of an autoregressive language modeling task.

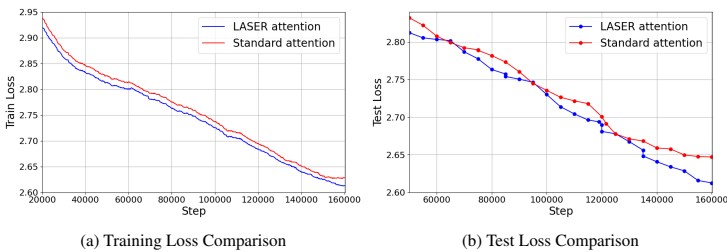

(a) Training Loss Comparison          (b) Test Loss Comparison

Figure 2: Comparison between LASER and Standard attention mechanism on a 301 million parameter autoregressive language model with 32 layers and 2048 hidden dimensions on C4 dataset for training and test loss vs steps for 167 billion tokens. LASER attention shows consistently lower loss.

**Dataset and setup.** We use the C4 dataset (Raffel et al., 2020) for our experiments. The training is conducted using a batch size of 1024 sequences, each with a sequence length of 1024 tokens. The models are trained for 160,000 iterations, resulting in the utilization of approximately 167.8 billion tokens. Throughout the training process, we monitor both the training and test losses, and we observe a significant improvement in the test set performance when using LASER Attention compared to the standard attention mechanism (as illustrated in Figure 2).

**Model architecture.** The base model architecture consists of 300 million parameters of a decoder-only Transformer, which is distributed across 32 layers. Each layer uses 8 attention heads, with each head having a size of 128. The MLP block in this architecture has a hidden dimension of 2048.

In addition to this configuration, we also experiment with a variant where the model retains 32 layers but increases the MLP block hidden dimension to 4096. In this variant, we increase the hidden dimension of the MLP block to shift more parameters into the MLP block. This configuration continues to show improvements in both the training and test loss metrics, demonstrating that the effectiveness of LASER Attention is maintained even when attention parameters are reduced. The results of these experiments can be seen in Table 1, where we also include ablation results showing improvements even with a 16-layer setting.

| Number of Layers | Hidden Dimension | LASER | Standard Attention |
|---|---|---|---|
| 16 | 4096 | **2.673** | 2.681 |
| 32 | 2048 | **2.595** | 2.641 |
| 32 | 4096 | **2.555** | 2.575 |

Table 1: Comparison of test loss between LASER and Standard attention mechanisms across different model configurations, where we notice upto 1.74% relative improvement in loss.

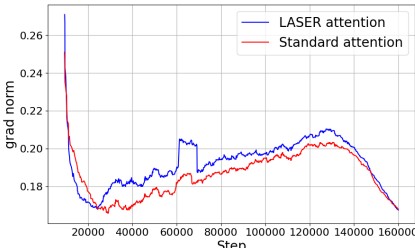

Figure 3: In this figure, we measure grad_norm vs steps for an autoregressive language model with a 301 million parameters model corresponding to Figure 2. Here we can notice that LASER attention has higher gradients throughout the training as discussed in Section 3.2.

**Ablation with optimizers.** In Figure 3, we observed that both gradient norms were higher compared to the baseline. An initial hypothesis was that higher gradient norms might lead to more parameter change, consequently reducing the loss more effectively. To investigate this, we utilized the LAMB optimizer (You et al., 2019), which normalizes and renormalizes updates using the weight norm to ensure that the scale of updates matches the scale of the weights, thus voiding the effect of gradient/update norms on optimization. Interestingly, even with LAMB's normalization mechanism, we observed a consistent improvement in training (Standard Attention - 2.749 vs LASER - 2.736) and test loss (Standard Attention - 2.758 vs LASER - 2.741), suggesting that the performance gains were not solely driven by larger gradient magnitudes but are intrinsic to the model's architecture and the LASER Attention mechanism.

**Scaling to larger models.** To demonstrate scalability of our approach, we conducted experiments on a 1.1 billion and 2.2 billion parameter model. Without the Log-Weighted-Sum-Exp trick we introduced in Section 3.3, we noticed that the 2.2 billion model training fails. In Figure 4, we show that LASER attention outperforms standard-attention in a 2.2 billion parameter model with model dimension 2048 and hidden dimension 8192 with 32 layers and 8 attention heads (each of size 512). We observe the same in 1.1 billion model (Figure 4), which has a scaled down hidden dimension (4096) and attention head size (256).

**Evaluation on downstream tasks.** We evaluate the performance of our 2.2 billion parameter model on several downstream tasks and mention in Table 2. Where we evaluate on ARC (Clark et al., 2018), BoolQ (Clark et al., 2019), CB (Wang et al., 2019), COPA (Wang et al., 2019), HellaSwag (Zellers et al., 2019), MultiRC (Khashabi et al., 2018), OpenBookQA (Mihaylov et al., 2018), PIQA (Bisk et al., 2020), RACE (Lai et al., 2017), ReCoRD (Zhang et al., 2018), RTE (Wang et al., 2019), StoryCloze (Mostafazadeh et al., 2016), WiC (Pilehvar & Camacho-Collados, 2019), Winograd (Levesque et al., 2012), Winogrande (Kocijan et al., 2020), and WSC (Wang et al., 2019). We found that LASER outperforms in 14 out 17 datasets with upto 3.38% difference and 0.85% difference on average in accuracy.

**Training and evaluation.** All experiments are conducted using the PAX framework (Research, 2023), built on JAX (Bradbury et al., 2018), and executed on TPUv5 chips (Cloud, 2023). We use 64 chips for 300 million parameter model, 128 chips for 1.1 billion and 256 chips for 2.2 billion parameter model. Each training run takes upto 24 hours. We conducted hyperparameter search on 16-layer model mentioned in Table 1 with 15 hyperparameters using search space in Table 4. We noticed that LASER attention exhibit fewer training curve spikes, which we note in Section A.3.

| (a) Part 1 | | |
| --- | --- | --- |
| **Dataset** | **LASER (mean±std)** | **Standard (mean±std)** |
| WSC | **81.12±0.41** | 79.23±0.41 |
| Winogrande | 62.04±0.21 | **62.26±0.14** |
| Winograd | **82.05±0.40** | 80.22±0.33 |
| WiC | **51.38±0.59** | 51.16±0.45 |
| StoryCloze | **77.97±0.10** | 76.42±0.08 |
| RTE | 53.07±0.23 | **53.29±0.29** |
| ReCoRD | **85.28±0.10** | 85.04±0.08 |
| RaceM | **50.56±0.12** | 49.69±0.10 |

| (b) Part 2 | | |
| --- | --- | --- |
| **Dataset** | **LASER (mean±std)** | **Standard (mean±std)** |
| RaceH | **37.82±0.12** | 37.58±0.16 |
| PIQA | **77.15±0.12** | 76.75±0.09 |
| OpenBookQA | **49.12±0.16** | 47.40±0.22 |
| MultiRC | **57.57±0.12** | 54.17±0.19 |
| HellaSwag | **66.62±0.05** | 65.46±0.03 |
| COPA | **82.00±0.00** | 80.20±0.40 |
| CB | 40.00±0.87 | **44.64±1.13** |
| BoolQ | **63.37±0.14** | 60.46±0.33 |

Table 2: Accuracies of one-shot evaluation of a 2.2 billion parameter autoregressive language model trained via LASER and standard attention. We found that LASER outperforms or performs the same as standard attention on 14 out of 17 datasets by up to 3.4%. On average, LASER gives an accuracy of 63.57±0.23% vs standard attention's 62.75±0.28%.

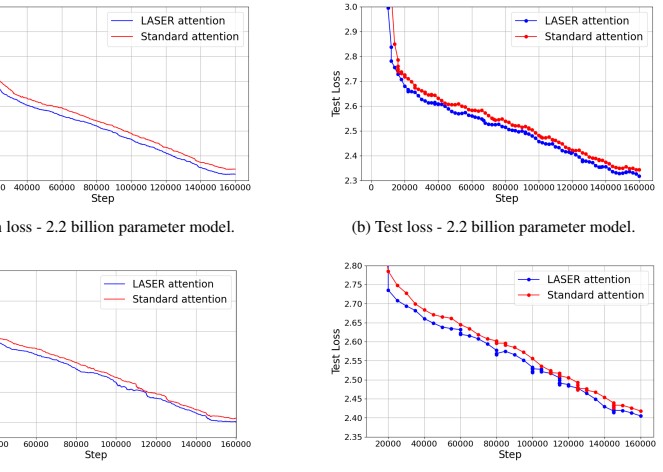

(a) Train loss - 2.2 billion parameter model.

(b) Test loss - 2.2 billion parameter model.

(c) Train loss - 1.1 billion parameter model.

(d) Test loss - 1.1 billion parameter model.

Figure 4: Performance comparison for 2.2 billion and 1.1 billion parameter models. The 2.2 billion model has 32 layers, 8 attention heads (head size 512), MLP hidden dimension 8192, and model dimension 2048. The 1.1 billion model has 32 layers, 8 attention heads (head size 256), MLP hidden dimension 4096, and model dimension 1024. We show that LASER outperforms Standard Attention in large scale settings.

## 4.2 MASKED LANGUAGE MODELING VIA BERT

In the experiments so far, the focus was mainly on decoder-only models, to diversify our evaluation we now shift to encoder-only model- BERT (Devlin et al., 2018) trained via masked language modeling (as opposed to next token prediction). We train a 2.2 billion parameter BERT on MLPerf training data which uses wikipedia articles. We get better error rate of masked language model predictions - LASER - 0.2125 vs Standard Attention - 0.2145 (0.93% relative improvement). One may notice that LASER makes more difference in decoder-only models compared to BERT. We used model dimension of 2048, hidden dimension - 8192, number of attention heads 16, each of size 256.

## 4.3 VISION TRANSFORMER (VIT) AND CONFORMER - SPEECH-TO-TEXT

**Vision Transformer (ViT) on Imagenet-1k.** In this section, we experiment with the Vision Transformer (ViT) S/16 (Dosovitskiy et al., 2021) variant on the Imagenet-1k classification task (Deng et al., 2009) which are part of AlgoPerf benchmarks (Dahl et al., 2023) for optimizer comparisons. These benchmarks are identically implemented in init2winit framework (Gilmer et al., 2023), build on JAX, which we use for our experiments.

A hyperparameter sweep was conducted over 50 configurations on NAdamW (Dozat, 2016), focusing on the search space defined in Table 3. We selected the best-performing hyperparameter configuration based on validation performance for standard attention, run it for 5 different random seeds (for initialization) and report the validation curves corresponding to median in Figure 5, where we

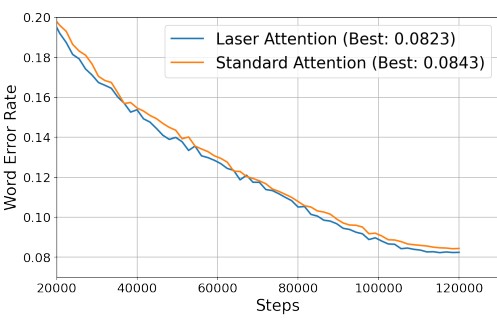 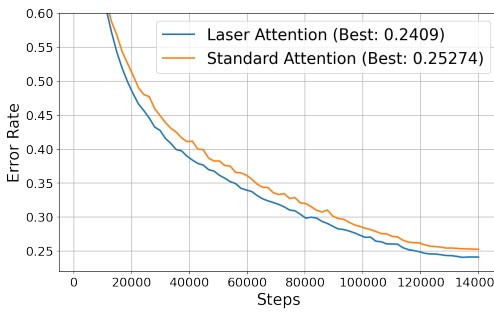

(a) Conformer Speech-to-Text - Validation     (b) ViT ImageNet Classification - Validation

Figure 5: Comparison of LASER attention vs Standard attention in two tasks: Conformer Speech-to-Text (left) and ViT ImageNet Classification (right). LASER attention provides a 1% absolute improvement in error rate (25.27% → 24.09%) i.e., a ∼4.67% relative improvement. In Conformer, we notice an improvement of word error rate (WER) (0.0843 → 0.0824) - 2.25% relative improvement. Median curve among 5 random initializations corresponding to the best performing hyperparameter configuration is reported.

show that LASER attention provides a 1% absolute improvement in error rate (25.27% → 24.12%), which translates to a ∼4% relative improvement over standard attention.

**Conformer on Librispeech Speech-to-Text.** We also evaluate the performance of LASER attention on the Librispeech Speech-to-Text dataset (Panayotov et al., 2015) using the Conformer model (Gulati et al., 2020). Similar to the ViT experiments, we use the AlgoPerf benchmark and perform a hyperparameter sweep across 50 configurations to optimize standard attention. We pick the optimal hyperparameters, run them for 5 different random seeds (for initialization) and report the validation curves corresponding to median in Figure 5 where we demonstrate a clear reduction in word error rate (WER) (0.0843 → 0.0824) when using LASER attention.

These experiments show that LASER attention improves both image classification and speech-to-text performance, further highlighting its versatility and efficiency across different modalities.

**Comparisons using OPTLists.** In AlgoPerf (Dahl et al., 2023), a list of 5 hyperparameters for NAdamW, tuned for a variety of benchmarks (as opposed to just Conformer and ViT) were provided - OPTList. We also evaluate our models on OPTList by running each hyperparameter with 5 different random seeds (initializations) and picking the median of the best performing hyperparameter. For Imagenet-ViT benchmark we obtain a reduction in validation error rate from 0.21732 to 0.21348. In Librispeech-Conformer benchmark we obtain a reduction validation word error rate from 0.07728 to 0.07607.

## 5 CONCLUSIONS

In this paper, we first identified a bottleneck in the gradient backpropagation of attention mechanism where the gradients are scaled by small Jacobian values while passing through the softmax operation. We fix this issue by transforming the inputs and outputs of attention mechanism, and show that this leads to larger Jacobians in the limiting case. We demonstrate the improvements in training performance over four types of transformers spanning different modalities (text, speech and vision): (a) decoder-only (via Large Language model) upto 2.2 billion parameters, (b) encoder-only (BERT) with 2.2 billion parameters, (c) vision Transformers on Imagenet, and (d) Conformer on Librispeech speech-to-text, where we show significant and consistent improvements in performance.

## 6 LIMITATIONS

While we conduct research on improving attention mechanism, which has wide applicability. Due to quadratic complexity in sequence length, scaling to large sequence lengths can be a major limitation of LASER or any attention mechanism.

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

# A APPENDIX

## A.1 HYPERPARAMETER SEARCH SPACE

In Table 3 we outline the hyperparameter search space for all the benchmarks in Section 4.3.

| Parameter | Min | Max | Scaling/Feasible Points |
|---|---|---|---|
| learning_rate | $10^{-4}$ | $10^{-2}$ | log |
| $1 - \beta_1$ | $10^{-2}$ | 0.15 | log |
| $\beta_2$ | - | - | 0.9, 0.99, 0.999 |
| warmup_factor | - | - | 0.05 |
| weight_decay | $5 \times 10^{-3}$ | 1.0 | log |
| label_smoothing | - | - | 0.1, 0.2 |
| dropout_rate | - | - | 0.1 |

Table 3: Hyperparameter search space used in Section 4.3.

| Parameter | Value |
|---|---|
| learning_rate | [1e-1, 1e-2, 1e-3, 1e-4, 1e-5] |
| weight_decay | [1e-2, 1e-1, 1.0] |
| beta_1 | 0.9 |
| beta_2 | 0.99 |
| epsilon | 1e-24 |
| dropout_rate | 0.0 |

Table 4: Hyperparameter search space for language modeling experiments, Section 4.1

## A.2 PROOFS

*Proof of Lemma 3.1.* The softmax activation function is applied row-wise on the preactivations $\tilde{A}$; we can expand this computation row-wise as follows:

$$A = \text{softmax}(\tilde{A})$$

$$\implies \begin{pmatrix} a_1^\top \\ \vdots \\ a_s^\top \end{pmatrix} = \begin{pmatrix} \text{softmax}(\tilde{a}_1^\top) \\ \vdots \\ \text{softmax}(\tilde{a}_s^\top) \end{pmatrix}$$

$$\implies a_i = \text{softmax}(\tilde{a}_i), \ \ i \in \{1, \ldots, N\}$$

$$= \left\{ \frac{\exp(\tilde{a}_{i1})}{\sum_k \exp(\tilde{a}_{ik})}, \ldots, \frac{\exp(\tilde{a}_{is})}{\sum_k \exp(\tilde{a}_{ik})} \right\}$$

$$\implies a_{ij} = \frac{\exp(\tilde{a}_{ij})}{\sum_k \exp(\tilde{a}_{ik})}$$

Taking gradient with respect to $\tilde{a}_i$ in the last expression gives:

$$\frac{\partial a_{ij}}{\partial \tilde{a}_{il}} = a_{ij}(1 - a_{ij}) \ \text{ if } l = j$$

$$= -a_{ij}a_{il} \ \text{ else}$$

Putting everything together, the Jacobian of the transformation $a_i = \text{softmax}(\tilde{a}_i)$ can be written as follows:

$$\frac{\partial a_{ij}}{\partial \tilde{a}_{il}} = (\text{diag}(a_i) - a_i a_i^\top)$$

$$\frac{\partial \ell}{\partial \tilde{a}_i} = (\text{diag}(a_i) - a_i a_i^\top) \frac{\partial \ell}{\partial a_i} \tag{8}$$

## A.3 TRAINING ANALYSIS

**Training instability.** There can be spikes in training curves initially during large language model training. We notice that despite these spikes training stabilizes and converges smoothly. However, training instability/spikes can be attributed to poor model architecture and optimizer choices. We now ablate the choice of attention mechanism and understand its affect on training stability. Figure 6 compares training stability of different models.

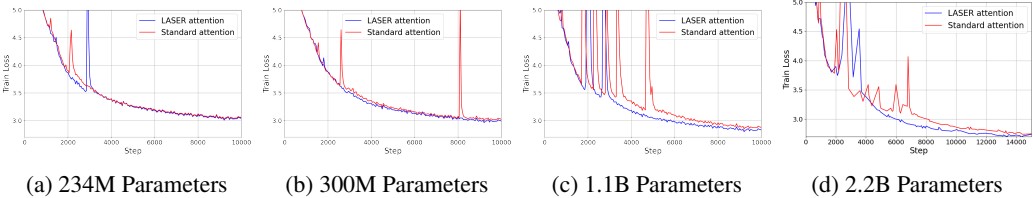

| (a) 234M Parameters | (b) 300M Parameters | (c) 1.1B Parameters | (d) 2.2B Parameters |

Figure 6: Train loss vs steps for LASER and standard attention across different number of parameters. The training stability for each attention mechanism can be observed through the number of training spikes. Generally, models with LASER attention exhibit fewer training spikes compared to models with standard attention, indicating greater stability in training for LASER attention across all parameter scales. We focus the figures on initial part of the training as the rest of the training didn't demonstrate any training instability.

| Model Size | LASER (hrs) | Standard Attention (hrs) | Overhead (hrs) | Percentage Overhead (%) |
|---|---|---|---|---|
| 234M | 12.08 | 11.61 | 0.47 | 3.80% |
| 300M | 19.53 | 19.05 | 0.48 | 2.40% |
| 1B | 25.99 | 25.17 | 0.82 | 3.27% |
| 2B | 28.04 | 27.48 | 0.56 | 2.04% |

Table 5: Comparison of walltimes (in hours) for LASER and Standard Attention across different language model sizes from Section 4.1. We note an overhead of 2-4 % compared to standard attention. However, our implementation is naive and the additional $\log(\cdot)$ and $\exp(\cdot)$ operations are not fused with the attention function.

**Scaling Law Analysis.** In Figure 7, we used a power law fit $f(n) = an^b$ to fit the final test loss values of autoregressive training runs in Section 4.1 as a function of number of parameters.

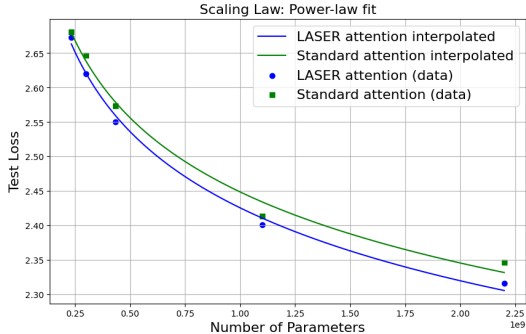

Figure 7: Scaling law: Power-law fit for test loss against number of parameters. This plot uses 234M, 300M, 435M, 1.1B and 2.2B parameter models' final test losses after training on $\sim$ 167B tokens. To reach a loss of 2.347, it takes 15.65% fewer parameters with LASER attention.

## A.4 LONG CONTEXT LANGUAGE MODELING

Long-context language modeling is an important research interest in large language modeling community. In this section, we evaluate LASER attention by scaling up context length to 8192 (in contrast to 1024 in Section 4.1) and model size to 5.2B parameters (in contrast to 2.2B in Section 4.1). We train this model on $\sim$ 40B tokens of Fineweb-Edu dataset (Lozhkov et al., 2024), where each sequence is of size 8192. The model uses 32 layers, hidden dimension of 7168 and model dimension of 4096. LASER reaches a training loss of 1.625 vs 1.632 reached by standard attention. We conduct evaluation on XSum (Narayan et al., 2018) and Scrolls-Qasper dataset (Shaham et al., 2022; Dasigi et al., 2021). In Table 6, we measure ther

| Context Length | LASER (Decoder F1) | Standard Attention (Decoder F1) |
|---|---|---|
| 2048 | **3.06** | 2.88 |
| 4096 | **3.11** | 2.48 |
| 8192 | **3.18** | 2.34 |

| Shots | ROUGE-1 | | ROUGE-Lsum | |
|---|---|---|---|---|
| | LASER | Standard | LASER | Standard |
| 5 | **8.95** | 8.70 | **7.38** | 6.90 |
| 10 | **8.95** | 8.34 | **7.43** | 6.59 |

Table 6: Comparison of LASER and Standard Attention for Scrolls-Qasper (Left) and XSum (Right).

SCROLLS-Qasper focuses on question answering over scientific research papers, requiring models to understand and synthesize information from long, complex documents. XSum, on the other hand, challenges models to generate concise, abstractive summaries of news articles, emphasizing informativeness and coherence.