# OpenReview forum: "LASER: Attention using Exponential Transformation"
_ICLR.cc/2025/Conference — Submitted to ICLR 2025_

### Official Review · Reviewer_CHtJ · 2024-11-03

**Soundness:** 3
**Presentation:** 3
**Contribution:** 2
**Rating:** 6
**Confidence:** 3

**Summary:**

This paper investigates the limitation of the softmax operation, which tends to backpropagate small gradients, potentially slowing down learning. To address this, the authors introduce a new attention primitive called LASER. LASER applies attention on exponentially transformed inputs, adopting a log-sum-exp structure. The authors demonstrate LASER's effectiveness through evaluations on language modeling, vision, and speech recognition tasks.

**Strengths:**

The paper is well-written, with a clearly motivated problem that addresses a meaningful limitation in current attention mechanisms. The proposed solution is notably simple, requiring minimal modifications to the original attention mechanism, allowing easy integration into existing system implementations. Additionally, the authors evaluate their new attention primitive across a broad and diverse set of tasks, effectively demonstrating its versatility and effectiveness.

**Weaknesses:**

My primary concern is the significance of the observed improvement. In Table 2, the average improvement of LASER over the standard attention mechanism across all benchmarks is less than 1%. It would be helpful to see the natural fluctuations on these benchmarks to better assess the statistical significance of this improvement. Additionally, although LASER shows a lower loss in the training curve, it is unclear if LASER merely offers faster convergence rather than a genuinely better final performance. To clarify this, the authors could extend the training schedule to verify that both models have indeed converged, demonstrating that LASER not only converges faster but also achieves a superior end performance.

While I hesitate to bring this up, a common question for any new attention mechanism is its scalability. The autoregressive language model used in this study is still below 3B parameters. It would strengthen the paper to show efforts towards scaling this approach (I realize this is easier said than done). Additionally, I’m curious if it might be possible to adapt an existing model trained with standard attention to LASER, which would reduce the computational burden of training a larger model from scratch.

Finally, while I understand the claim that existing system implementations for standard attention can be reused, I would still like to know if any additional overhead is introduced. Presumably, log and exp operations could be fused into the kernel, but it would be helpful to see specific performance metrics to quantify any potential overhead.

**Questions:**

Please address the questions raised in the weaknesses section.

---

> ### Author Response · Authors · 2024-11-25
> **Fluctuations in Downstream numbers,  Larger model numbers, Walltime overhead**
>
> We thank the reviewer for the detailed review.
>
> ---
>
> > My primary concern is the significance of the observed improvement. In Table 2, the average improvement of LASER over the standard attention mechanism across all benchmarks is less than 1%. It would be helpful to see the natural fluctuations on these benchmarks to better assess the statistical significance of this improvement.
>
> We measured fluctuations across 5 checkpoints for both LASER and standard attention and updated Table 2. We note a difference of 0.82 in accuracy: LASER 63.57$\pm$0.23 $\%$,  Standard attention 62.75$\pm$0.28 $\%$. We note that the difference is much higher compared to standard deviation.
>
> ## Part 1: LASER vs Standard
>
> | Dataset     | LASER (mean±std)  | Standard (mean±std) |
> |-------------|-------------------|---------------------|
> | WSC         | **81.12±0.41**    | 79.23±0.41          |
> | Winogrande  | 62.04±0.21        | **62.26±0.14**      |
> | Winograd    | **82.05±0.40**    | 80.22±0.33          |
> | WiC         | **51.38±0.59**    | 51.16±0.45          |
> | StoryCloze  | **77.97±0.10**    | 76.42±0.08          |
> | RTE         | 53.07±0.23        | **53.29±0.29**      |
> | ReCoRD      | **85.28±0.10**    | 85.04±0.08          |
> | RaceM       | **50.56±0.12**    | 49.69±0.10          |
>
> ## Part 2: LASER vs Standard
>
> | Dataset     | LASER (mean±std)  | Standard (mean±std) |
> |-------------|-------------------|---------------------|
> | RaceH       | **37.82±0.12**    | 37.58±0.16          |
> | PIQA        | **77.15±0.12**    | 76.75±0.09          |
> | OpenBookQA  | **49.12±0.16**    | 47.40±0.22          |
> | MultiRC     | **57.57±0.12**    | 54.17±0.19          |
> | HellaSwag   | **66.62±0.05**    | 65.46±0.03          |
> | COPA        | **82.00±0.00**    | 80.20±0.40          |
> | CB          | 40.00±0.87        | **44.64±1.13**      |
> | BoolQ       | **63.37±0.14**    | 60.46±0.33          |
>
>
>
> ----
>
> > While I hesitate to bring this up, a common question for any new attention mechanism is its scalability. The autoregressive language model used in this study is still below 3B parameters. It would strengthen the paper to show efforts towards scaling this approach (I realize this is easier said than done).
>
> We scaled up to 5.2 billion parameters and context length of 8192 and found improvements of LASER over standard attention. We also included scaling laws analysis in Section A.3, which shows a steeper curve for LASER. Long-context language modeling is an important research interest in the large language modeling community. We evaluate LASER attention by **scaling up context length to 8192 (in contrast to 1024 in Section 4.1) and model size to 5.2B parameters (in contrast to 2.2B in Section 4.1)**. We train this model on ~40B tokens of the Fineweb-Edu dataset ([Lozhkov et al., 2024](#lozhkov2024finewebedu)). The model uses 32 layers, a hidden dimension of 7168, and a model dimension of 4096. LASER reaches a training loss of **1.625** versus **1.632** reached by standard attention.
>
> We conduct evaluation on the XSum ([Narayan et al., 2018](#narayanetal2018dont)) and Scrolls-Qasper datasets ([Shaham et al., 2022; Dasigi et al., 2021](#shaham2022scrolls)). In the table below, we compare their performance.
>
> | Context Length   | LASER (Decoder F1) | Standard Attention (Decoder F1) |
> |------------------|--------------------|----------------------------------|
> | **2048**         | **3.06**          | 2.88                             |
> | **4096**         | **3.11**          | 2.48                             |
> | **8192**         | **3.18**          | 2.34                             |
>
> | Shots   | ROUGE-1 (LASER) | ROUGE-1 (Standard) | ROUGE-Lsum (LASER) | ROUGE-Lsum (Standard) |
> |---------|------------------|--------------------|---------------------|-----------------------|
> | **5**   | **8.95**         | 8.70               | **7.38**            | 6.90                 |
> | **10**  | **8.95**         | 8.34               | **7.43**            | 6.59                 |
>
> *Table: Comparison of LASER and Standard Attention for Scrolls-Qasper (Left) and XSum (Right). LASER outperforms in both XSum and Scrolls-Qasper*

---

> > ### Author Response · Authors · 2024-11-25
> > **Walltime overhead**
> >
> > ----
> > > Finally, while I understand the claim that existing system implementations for standard attention can be reused, I would still like to know if any additional overhead is introduced. Presumably, log and exp operations could be fused into the kernel, but it would be helpful to see specific performance metrics to quantify any potential overhead.
> >
> >
> >
> > We noticed that LASER is slightly slower than standard softmax attention, but we noticed that for models with higher hidden dimensions (2.2B in the table), the latency reduces. We conduct wall-time analysis of our experiments on language models in Section 4.1, we mention this in Section A.3 (Training Analysis).  We will include wall-time of inference phase in our final draft.
> >
> >
> > ## Wall-time Analysis
> >
> > | **Model Size** | **LASER (hrs)** | **Standard Attention (hrs)** | **Overhead (hrs)** | **Percentage Overhead (%)** |
> > |----------------|------------------|------------------------------|--------------------|-----------------------------|
> > | **234M**       | 12.08           | 11.61                       | 0.47               | 3.80%                       |
> > | **300M**       | 19.53           | 19.05                       | 0.48               | 2.40%                       |
> > | **1.1B**         | 25.99           | 25.17                       | 0.82               | 3.27%                       |
> > | **2.2B**         | 28.04           | 27.48                       | 0.56               | 2.04%                       |
> >
> > *Table: Comparison of walltimes (in hours) for LASER and Standard Attention across different language model sizes. We note an overhead of 2-4% compared to standard attention. However, the implementation is naive, and the additional $\log(\cdot)$ and $\exp(\cdot)$ operations are not fused with the attention function.*

---

> > > ### Comment · Reviewer_CHtJ · 2024-11-25
> > >
> > > Thank you for your detailed response. I have increased my score by 1. I highly encourage you to incorporate these new findings into your revisions and improve the efficiency of the design using the fused kernel approach.

---

> > > > ### Author Response · Authors · 2024-11-25
> > > >
> > > > Thank you for your consideration and re-evaluation of the score. We are excited to implement a fused kernel to improve LASER's performance.

---

### Official Review · Reviewer_Asoj · 2024-11-03

**Soundness:** 2
**Presentation:** 3
**Contribution:** 2
**Rating:** 5
**Confidence:** 4

**Summary:**

This paper introduces LASER attention, a new attention mechanism designed to improve the gradient propagation in Transformers by replacing the standard softmax-based attention. The softmax operation in traditional attention mechanisms can limit learning due to small gradient backpropagation, while LASER attention uses a log-sum-exp structure to allow larger gradient signals, enhancing model training.

**Strengths:**

1. The motivation of this paper is insightful.
2. LASER attention is straightforward to implement, requiring minimal adjustments to current attention models.
3. The experiments are quite comprehensive, verifying the effectiveness of LASER across different modalities.

**Weaknesses:**

My only concern is whether this paper addresses **an issue that standard attention cannot resolve**. If standard attention is suboptimal due to small gradient backpropagation, this could potentially be improved by adjusting the temperature of the softmax (i.e., scaling its input). I suggest that the paper examine the impact of temperature both theoretically and experimentally.

**Questions:**

Please incorporate the temperature of softmax into the theoretical derivations and comparative experiments.

---

> ### Author Response · Authors · 2024-11-25
> **Impact of Temperature**
>
> We thank you for raising an important issue.  We used temperature as $\sqrt{d}$ as default in our experiments, where $d$ denotes the size of the attention head, as is the default in Vaswani et al., 2017. Here are the formulation of attention and LASER attention with temperature included:
> $$ \text{attn}(Q,K,V,\tau) = \text{softmax}(QK^T/\tau)V,\ \tau=\sqrt{d},$$
> $$\text{laser}(Q,K,V,\tau) = \log(\text{softmax}(QK^T/\tau)\exp(V)),\ \tau=\sqrt{d}.$$
>
> To understand the impact of temperature, we tuned both LASER and standard attention with $\tau\in\{\sqrt{d},2\sqrt{d},16\sqrt{d}\}$, for ViT experiment in Section 4.3 and still found an improvement of LASER over standard attention. Here's the new result: 24.24 (Standard) vs 24.0 (LASER).  We recognize the impact of temperature on Softmax, since both standard attention and LASER's numbers improve compared to the previous numbers, however the aforementioned result indicates that LASER still has a complementary improvement on performance that tuning temperature doesn't address. Due to time constraints we couldn't do the same for more of our experiments and conduct theoretical analysis. We will try to address that in our final revision.
>
> 1. Vaswani, A. "Attention is all you need." Advances in Neural Information Processing Systems (2017).

---

> > ### Author Response · Authors · 2024-11-25
> > **Gentle reminder of the discussion deadline**
> >
> > Thank you once again for taking the time to review our manuscript. We understand and appreciate your busy schedule, and we would like to kindly remind you that the discussion deadline is approaching. If there are any outstanding concerns or suggestions for further improving our manuscript, we would be grateful to receive your feedback. Additionally, if you believe we have adequately addressed the points raised in your initial review, we would sincerely appreciate your consideration in revisiting the score assigned to the paper.

---

> ### Comment · Reviewer_Asoj · 2024-11-26
>
> From the theory in the paper, the suboptimality of standard attention arises because the term $\sigma(\tilde{a}\_{11} - \tilde{a}\_{12})(1 - \sigma(\tilde{a}\_{11} - \tilde{a}\_{12}))$ in the Jacobian tends to saturate when $\tilde{a}\_{11} - \tilde{a}\_{12}$ becomes too large. However, by introducing temperature into this theory, the term becomes $\sigma\left(\frac{\tilde{a}\_{11} - \tilde{a}\_{12}}{T}\right)\left(1 - \sigma\left(\frac{\tilde{a}\_{11} - \tilde{a}\_{12}}{T}\right)\right)$. Therefore, when $T$ is large, it can reduce $(\tilde{a}\_{11} - \tilde{a}\_{12})/T$, alleviating saturation.
>
>
> The authors present experimental results on ViT, which demonstrate that by adjusting the softmax temperature, the performance of standard attention improves by 1% (from 25.27% to 24.24%), while the performance of LASER attention remains almost unchanged (from 24.09% to 24.0%). Although the authors claim that the 0.24% performance gap still reflects the superiority of LASER attention, I believe this at least suggests that, through careful adjustment of the softmax temperature, standard attention can achieve similar performance to LASER attention. Therefore, although rebuttal time is limited, I still suggest that the authors thoroughly evaluate the impact of the softmax temperature across all experiments and include "standard attention + temperature tuning"  as a baseline  in future improvements.

---

> ### Author Response · Authors · 2024-11-28
>
> We acknowledge the improvement in standard attention for ViT benchmark and thank the reviewer for pointing this out. Our main experiment of the paper is the 2.2B parameter large language model, so we pretrained a temperature tuned baseline from scratch (picking temperature with best final test loss) added it to the downstream evals (mean and standard deviation over multiple checkpoints):
>
> ## Part 1: LASER vs Standard vs Standard+Temp Tuned
>
> | Dataset     | LASER   | Standard  | Standard+Temp Tuned  |
> |-------------|-------------------|---------------------|--------------------------------|
> | WSC         | **81.12±0.41**    | 79.23±0.41          | 78.25±0.22                     |
> | Winogrande  | 62.04±0.21        | **62.26±0.14**      | 62.19±0.33                     |
> | Winograd    | **82.05±0.40**    | 80.22±0.33          | 79.85±0.40                     |
> | WiC         | **51.38±0.59**    | 51.16±0.45          | 47.18±0.00                     |
> | StoryCloze  | **77.97±0.10**    | 76.42±0.08          | 76.17±0.15                     |
> | RTE         | 53.07±0.23        | **53.29±0.29**      | 51.70±0.27                     |
> | ReCoRD      | **85.28±0.10**    | 85.04±0.08          | 84.89±0.07                     |
> | RaceM       | **50.56±0.12**    | 49.69±0.10          | 48.89±0.09                     |
>
> ## Part 2: LASER vs Standard vs Standard+Temp Tuned
>
> | Dataset     | LASER  | Standard  | Standard+Temp Tuned  |
> |-------------|-------------------|---------------------|--------------------------------|
> | RaceH       | **37.82±0.12**    | 37.58±0.16          | 37.04±0.09                     |
> | PIQA        | **77.15±0.12**    | 76.75±0.09          | 76.45±0.11                     |
> | OpenBookQA  | **49.12±0.16**    | 47.40±0.22          | 48.48±0.30                     |
> | MultiRC     | **57.57±0.12**    | 54.17±0.19          | 53.38±0.16                     |
> | HellaSwag   | **66.62±0.05**    | 65.46±0.03          | 65.84±0.08                     |
> | COPA        | **82.00±0.00**    | 80.20±0.40          | **85.00±0.00**                 |
> | CB          | 40.00±0.87        | **44.64±1.13**      | 42.86±0.00                     |
> | BoolQ       | **63.37±0.14**    | 60.46±0.33          | 61.69±0.16                     |
>
> **We note no significant differences observed on standard attention's average performance upon tuning temperature parameter: LASER (63.57$\pm$0.23\%), Standard Attention (62.75$\pm$0.28\%), and temperature tuned standard attention (62.49$\pm$0.15\%).**
>
> We have made modifications to the paper and submitted a revised version. In our final revision, we plan to include updated ViT results along with other revised results. However, due to time constraints, we were unable to incorporate the temperature-tuned baseline for all experiments. We kindly request the reviewer to reconsider their score in light of these updates.

---

> > ### Author Response · Authors · 2024-12-01
> > **Gentle Reminder**
> >
> > We understand and appreciate your busy schedule, and we would like to kindly remind you that the discussion deadline is approaching. If there are any outstanding concerns or suggestions for further improving our manuscript, we would be grateful to receive your feedback. Additionally, if you believe we have adequately addressed the points raised recently, we would sincerely appreciate your consideration in revisiting the score assigned to the paper.

---

> > > ### Author Response · Authors · 2024-12-02
> > > **Gentle Reminder**
> > >
> > > We would like to kindly remind you that the discussion deadline is approaching. If there are any outstanding concerns or suggestions for further improving our manuscript, we would be grateful to receive your feedback. Additionally, if you believe we have adequately addressed the points raised recently, we would sincerely appreciate your consideration in revisiting the score assigned to the paper.

---

### Official Review · Reviewer_kv6R · 2024-11-05

**Soundness:** 3
**Presentation:** 2
**Contribution:** 2
**Rating:** 5
**Confidence:** 4

**Summary:**

This paper addresses the gradient vanishing issue in transformer models, a challenge that limits the effectiveness of deep learning in capturing long-range dependencies. To tackle this, the authors introduce LASER attention, a novel adjustment to the attention mechanism that uses a log-sum-exp transformation on exponentially scaled inputs to improve gradient propagation. This approach avoids gradient vanishing more effectively than traditional softmax-based attention mechanisms. The authors provide a new implementation, Weighted-Sum-Exp trick, to prevent overflow issues and demonstrate that LASER attention improves model performance across various transformer architectures and tasks. Empirical results show notable gains in accuracy and reduction in error rates in speech, vision, and language models, making LASER attention a feasible and efficient alternative for large-scale transformer applications.

**Strengths:**

1. Author present a modified attention mechanism for transformers that addresses the gradient vanishing issue by utilizing a log-sum-exp transformation.
2. LASER presented algorithm enhances gradient propagation without the need for complex changes to existing architectures.

**Weaknesses:**

1.	Evaluation in Table 2 of the LLM can be expanded to include retrieval and generation tasks, such as those demonstrated in Scrolls (see: https://arxiv.org/pdf/2201.03533) and Needle In A Haystack (see: https://github.com/gkamradt/LLMTest_NeedleInAHaystack).
2..	It would be interesting to know if LASER is still compatible with LoRA.
3.	A speed comparison between LASER and vanilla attention during both training and inference phases would be helpful.
4.	The author operates decoder-only causal language models ranging from 234 million to 2.2 billion parameters. A comparison of loss across all these models would help demonstrate LASER’s scalability from smaller to larger models.

**Questions:**

Beyond the weakness, the presentation of table and chart be imporved.

---

> ### Author Response · Authors · 2024-11-24
> **Long Context Results, Wall-time Analysis**
>
> Thank you for your review. We conducted additional long-context experiments, speed comparisons, and scaling laws to address your concerns.
>
> ---
>
> > Evaluation in Table 2 of the LLM can be expanded to include retrieval and generation tasks, such as those demonstrated in Scrolls (see: https://arxiv.org/pdf/2201.03533) and Needle In A Haystack (see: https://github.com/gkamradt/LLMTest_NeedleInAHaystack). 2.. It would be interesting to know if LASER is still compatible with LoRA.
>
> We included results on Xsum and Scrolls-Qasper datasets. Scrolls-Qasper focuses on question answering over scientific research papers, requiring models to understand and synthesize information from long, complex documents. XSum, on the other hand, challenges models to generate concise, abstractive summaries of news articles, emphasizing informativeness and coherence. Both tasks involve using context/prefix with size upto 8192 and to generate answers in our evaluation, thus making it a challenging long context retrieval and generation task. We summarize these results in Section A.4 and also repeat it here:
>
> ## Long Context Language Modeling
>
> Long-context language modeling is an important research interest in the large language modeling community. We evaluate LASER attention by **scaling up context length to 8192 (in contrast to 1024 in Section 4.1) and model size to 5.2B parameters (in contrast to 2.2B in Section 4.1)**. We train this model on ~40B tokens of the Fineweb-Edu dataset ([Lozhkov et al., 2024](#lozhkov2024finewebedu)). The model uses 32 layers, a hidden dimension of 7168, and a model dimension of 4096. LASER reaches a training loss of **1.625** versus **1.632** reached by standard attention.
>
> We conduct evaluation on the XSum ([Narayan et al., 2018](#narayanetal2018dont)) and Scrolls-Qasper datasets ([Shaham et al., 2022; Dasigi et al., 2021](#shaham2022scrolls)). In the table below, we compare their performance.
>
> | Context Length   | LASER (Decoder F1) | Standard Attention (Decoder F1) |
> |------------------|--------------------|----------------------------------|
> | **2048**         | **3.06**          | 2.88                             |
> | **4096**         | **3.11**          | 2.48                             |
> | **8192**         | **3.18**          | 2.34                             |
>
> | Shots   | ROUGE-1 (LASER) | ROUGE-1 (Standard) | ROUGE-Lsum (LASER) | ROUGE-Lsum (Standard) |
> |---------|------------------|--------------------|---------------------|-----------------------|
> | **5**   | **8.95**         | 8.70               | **7.38**            | 6.90                 |
> | **10**  | **8.95**         | 8.34               | **7.43**            | 6.59                 |
>
> *Table: Comparison of LASER and Standard Attention for Scrolls-Qasper (Left) and XSum (Right). LASER outperforms in both XSum and Scrolls-Qasper*
>
> We will include needle in haystack results and LoRA compatibility in our final draft, we unfortunately couldn't conduct these experiments due to time constraints.
>
> ----
>
> >  A speed comparison between LASER and vanilla attention during both training and inference phases would be helpful.
>
> We noticed that LASER is slightly slower than standard softmax attention, but we noticed that for models with higher hidden dimensions (2.2B in the table), the latency reduces. We conduct wall-time analysis of our experiments on language models in Section 4.1, we mention this in Section A.3 (Training Analysis).  We will include wall-time of inference phase in our final draft.
>
>
> ## Wall-time Analysis
>
> | **Model Size** | **LASER (hrs)** | **Standard Attention (hrs)** | **Overhead (hrs)** | **Percentage Overhead (%)** |
> |----------------|------------------|------------------------------|--------------------|-----------------------------|
> | **234M**       | 12.08           | 11.61                       | 0.47               | 3.80%                       |
> | **300M**       | 19.53           | 19.05                       | 0.48               | 2.40%                       |
> | **1.1B**         | 25.99           | 25.17                       | 0.82               | 3.27%                       |
> | **2.2B**         | 28.04           | 27.48                       | 0.56               | 2.04%                       |
>
> *Table: Comparison of walltimes (in hours) for LASER and Standard Attention across different language model sizes. We note an overhead of 2-4% compared to standard attention. However, the implementation is naive, and the additional $\log(\cdot)$ and $\exp(\cdot)$ operations are not fused with the attention function.*
>
>
> ---
>
> ### References
>
> - Lozhkov et al., 2024. *Fineweb-Edu dataset*.
> - Narayan et al., 2018. *XSum Dataset*.
> - Shaham et al., 2022; Dasigi et al., 2021. *SCROLLS-Qasper Dataset*.

---

> ### Author Response · Authors · 2024-11-24
> **Scaling law analysis**
>
> ----
>
> >  The author operates decoder-only causal language models ranging from 234 million to 2.2 billion parameters. A comparison of loss across all these models would help demonstrate LASER’s scalability from smaller to larger models.
>
> We derived scaling laws for LASER and Standard Attention and found steeper curve for LASER. Please refer to Scaling Laws Analysis in Section A.3 (Training Analysis).
>
> ---
>
> > Beyond the weakness, the presentation of table and chart be imporved.
>
> We improved the presentation of Table 2 to make it more concise.

---

> > ### Author Response · Authors · 2024-11-25
> > **Gentle reminder of the discussion deadline**
> >
> > Thank you once again for taking the time to review our manuscript. We understand and appreciate your busy schedule, and we would like to kindly remind you that the discussion deadline is approaching. If there are any outstanding concerns or suggestions for further improving our manuscript, we would be grateful to receive your feedback. Additionally, if you believe we have adequately addressed the points raised in your initial review, we would sincerely appreciate your consideration in revisiting the score assigned to the paper.

---

> > > ### Author Response · Authors · 2024-11-26
> > > **Gentle Reminder of Discussion Deadline**
> > >
> > > We understand and appreciate your busy schedule, and we would like to kindly remind you that the discussion deadline is approaching. If there are any outstanding concerns or suggestions for further improving our manuscript, we would be grateful to receive your feedback. Additionally, if you believe we have adequately addressed the points raised in your initial review, we would sincerely appreciate your consideration in revisiting the score assigned to the paper.

---

> > > > ### Author Response · Authors · 2024-12-01
> > > > **Gentle reminder**
> > > >
> > > > We understand and appreciate your busy schedule, and we would like to kindly remind you that the discussion deadline is approaching. If there are any outstanding concerns or suggestions for further improving our manuscript, we would be grateful to receive your feedback. Additionally, if you believe we have adequately addressed the points raised in your initial review, we would sincerely appreciate your consideration in revisiting the score assigned to the paper.

---

> > > > > ### Author Response · Authors · 2024-12-02
> > > > > **Gentle reminder**
> > > > >
> > > > > We would like to kindly remind you that the discussion deadline is approaching. If there are any outstanding concerns or suggestions for further improving our manuscript, we would be grateful to receive your feedback. Additionally, if you believe we have adequately addressed the points raised recently, we would sincerely appreciate your consideration in revisiting the score assigned to the paper.

---

### Official Review · Reviewer_Qigg · 2024-11-09

**Soundness:** 3
**Presentation:** 3
**Contribution:** 3
**Rating:** 6
**Confidence:** 4

**Summary:**

This paper first analyzed the gradient vanishing problem in the standard softmax attention. To mitigate this issue, this paper proposed LASER attention, which first apply exponential function to the values in attention then take the log of the attention output. The authors clear explain why LASER attention mitigate the small gradient issue in theory.

Experimentally, the authors reported results in language modeling, image classification on ImageNet and speech-to-text generation. LASER attention achieved improvements over standard softmax attention on all these tasks.

**Strengths:**

1. The proposed LASER attention is theoretically grounded.

2. The authors provided an simple implementation of LASER attention by leveraging the log-sum-exp operation.

3. The experimental results demonstrates the effectiveness of LASER attention.

4. The paper is well-written, easy to follow.

**Weaknesses:**

1. In auto-regressive language modeling experiments, all the models are trained with context length 1024. No experimental results are reported for long-context training and evaluation.

2. When designing a new architecture, training stability is an important factor in consideration. However, there are no analysis in this paper to compare the training stability of LASER and the standard softmax attention.

**Questions:**

1. Are the language models trained using precision BF16 or FP32? Is LASER attention more stable (less training spikes)  or not?

2. Is LASER attention slower than standard softmax attention. Any efficiency analysis?

---

> ### Author Response · Authors · 2024-11-24
> **Included Long Context Experiments, Training Stability**
>
> Thanks for your critical review. We are delighted to know that you found our work easy to follow. We conducted experiments to address your concerns.
>
> ---
>
> > In auto-regressive language modeling experiments, all the models are trained with context length 1024. No experimental results are reported for long-context training and evaluation.
>
>
> ## Long Context Language Modeling
>
> Long-context language modeling is an important research interest in the large language modeling community. We evaluate LASER attention by **scaling up context length to 8192 (in contrast to 1024 in Section 4.1) and model size to 5.2B parameters (in contrast to 2.2B in Section 4.1)**. We train this model on ~40B tokens of the Fineweb-Edu dataset ([Lozhkov et al., 2024](#lozhkov2024finewebedu)). The model uses 32 layers, a hidden dimension of 7168, and a model dimension of 4096. LASER reaches a training loss of **1.625** versus **1.632** reached by standard attention.
>
> We conduct evaluation on the XSum ([Narayan et al., 2018](#narayanetal2018dont)) and Scrolls-Qasper datasets ([Shaham et al., 2022; Dasigi et al., 2021](#shaham2022scrolls)). In the table below, we compare their performance.
>
> | Context Length   | LASER (Decoder F1) | Standard Attention (Decoder F1) |
> |------------------|--------------------|----------------------------------|
> | **2048**         | **3.06**          | 2.88                             |
> | **4096**         | **3.11**          | 2.48                             |
> | **8192**         | **3.18**          | 2.34                             |
>
> | Shots   | ROUGE-1 (LASER) | ROUGE-1 (Standard) | ROUGE-Lsum (LASER) | ROUGE-Lsum (Standard) |
> |---------|------------------|--------------------|---------------------|-----------------------|
> | **5**   | **8.95**         | 8.70               | **7.38**            | 6.90                 |
> | **10**  | **8.95**         | 8.34               | **7.43**            | 6.59                 |
>
> *Table: Comparison of LASER and Standard Attention for Scrolls-Qasper (Left) and XSum (Right). LASER outperforms in both XSum and Scrolls-Qasper*
>
>
> SCROLLS-Qasper focuses on question answering over scientific research papers, requiring models to understand and synthesize information from long, complex documents. XSum, on the other hand, challenges models to generate concise, abstractive summaries of news articles, emphasizing informativeness and coherence.
>
> ----
> > When designing a new architecture, training stability is an important factor in consideration. However, there are no analysis in this paper to compare the training stability of LASER and the standard softmax attention.
>
>
> ## Training Instability
>
> There can be spikes in training curves initially during large language model training. We notice that despite these spikes, training stabilizes and converges smoothly. However, training instability/spikes can be attributed to poor model architecture and optimizer choices. We ablate the choice of attention mechanism and understand its effect on training stability and mention these observations in Section A.3 (Training Instability) Figure 6, which compares the training stability of different models. Training stability for each attention mechanism is observed through the number of training spikes. **Models with LASER attention surprisingly exhibit fewer spikes, indicating greater stability across all parameter scales. The analysis focuses on the initial part of training, as the rest of the training did not demonstrate significant instability.**
>
> ---
> > Are the language models trained using precision BF16 or FP32?
>
> We conduct our experiments in FP32 precision, to ensure accurate comparison. We will include experiments on BF16 in our final draft.
>
> ---
>
>
>
> ### References
>
> - Lozhkov et al., 2024. *Fineweb-Edu dataset*.
> - Narayan et al., 2018. *XSum Dataset*.
> - Shaham et al., 2022; Dasigi et al., 2021. *SCROLLS-Qasper Dataset*.

---

> > ### Author Response · Authors · 2024-11-24
> > **Wall-time Analysis**
> >
> > > Is LASER attention slower than standard softmax attention. Any efficiency analysis?
> >
> > We noticed that LASER is slightly slower than standard softmax attention, but we noticed that for models with higher hidden dimensions (2.2B in the table), the latency reduces. We conduct wall-time analysis of our experiments on language models in Section 4.1, we mention this in Section A.3 (Training Analysis).
> >
> >
> > ## Wall-time Analysis
> >
> > | **Model Size** | **LASER (hrs)** | **Standard Attention (hrs)** | **Overhead (hrs)** | **Percentage Overhead (%)** |
> > |----------------|------------------|------------------------------|--------------------|-----------------------------|
> > | **234M**       | 12.08           | 11.61                       | 0.47               | 3.80%                       |
> > | **300M**       | 19.53           | 19.05                       | 0.48               | 2.40%                       |
> > | **1.1B**         | 25.99           | 25.17                       | 0.82               | 3.27%                       |
> > | **2.2B**         | 28.04           | 27.48                       | 0.56               | 2.04%                       |
> >
> > *Table: Comparison of walltimes (in hours) for LASER and Standard Attention across different language model sizes. We note an overhead of 2-4% compared to standard attention. However, the implementation is naive, and the additional $\log(\cdot)$ and $\exp(\cdot)$ operations are not fused with the attention function.*
> >
> > ---

---

> > > ### Author Response · Authors · 2024-11-25
> > > **Gentle reminder of the discussion deadline**
> > >
> > > Thank you once again for taking the time to review our manuscript. We understand and appreciate your busy schedule, and we would like to kindly remind you that the discussion deadline is approaching. If there are any outstanding concerns or suggestions for further improving our manuscript, we would be grateful to receive your feedback. Additionally, if you believe we have adequately addressed the points raised in your initial review, we would sincerely appreciate your consideration in revisiting the score assigned to the paper.

---

> > ### Comment · Reviewer_Qigg · 2024-11-26
> > **Re: Official Comments**
> >
> > Really appreciated the great efforts for providing new experiments and analysis. The analysis on training instability and wall-time are convincing and address most of my questions. I highly recommend to conduct experiments in half precision (i.e. BF16) to better analyze training instability.
> >
> > For long-context experiments, the problem is that the results are too low to demonstrate convincing effectiveness. F1 score at around 3 or ROUGE score around 8 are nearly random.
> >
> > Overall, this is a good paper and I prefer to a recommendation of `accept`.

---

> > > ### Author Response · Authors · 2024-11-28
> > >
> > > >  I highly recommend to conduct experiments in half precision (i.e. BF16) to better analyze training instability.
> > >
> > > We conducted a pretraining experiment with 2.2 billion parameter except in BF16 activations and model parameters. Here's the we put the final training and test loss alongside the FP32 training.
> > >
> > > | Precision | LASER Loss | Standard Attention Loss |
> > > |-----------|------------|--------------------------|
> > > | FP32      | 2.326      | 2.344                   |
> > > | BF16      | 2.333      | 2.350                   |
> > >
> > > > For long-context experiments, the problem is that the results are too low to demonstrate convincing effectiveness. F1 score at around 3 or ROUGE score around 8 are nearly random.
> > >
> > > We acknowledge that the absolute f1 and ROUGE scores are low, however in the table below we note that the loss achieved by LASER is superior to that of Standard attention. We added another long-context experiment run by varying batchsize and observe significant differences in loss value. We will train for more tokens for our final revision with higher absolute f1 and rouge scores.
> > >
> > >  | Setting                                   | LASER   | Standard Attention |
> > > |-------------------------------------------|---------|--------------------|
> > > | 55 billion tokens, 256 batch size, 26k iterations  | 1.445   | 1.473              |
> > > | 47 billion tokens, 1024 batch size, 5.6k iterations | 1.613   | 1.622              |

---

### Meta-Review · Area_Chair_ASE6 · 2024-12-20

**Metareview:**

This paper introduces a modification to the attention mechanism coined LASER. This mechanism is supposed to improve gradient propagation in transformers. There are many concerns with this work: 1) the empirical validation is not very thorough and lacks comparison points. The tables are mostly empty of baselines. 2) it is not clear if it would not be possible to obtain similar gains by simply tuning the temperature parameter in the softmax. 3) The computation / performance tradeoff is not discussed properly. Overall, the paper's weaknesses outweigh the contribtions and the the work feels more like a workshop paper. I recommend this paper for rejection for ICLR 2025.

**Additional Comments On Reviewer Discussion:**

The authors have engaged with reviewers in some discussion, but the latter did not manage to convince the reviewers to strongly champion the paper for acceptance.

---

### Decision · Program_Chairs · 2025-01-22

Reject